# RNA Granules: A View from the RNA Perspective

**DOI:** 10.3390/molecules25143130

**Published:** 2020-07-08

**Authors:** Siran Tian, Harrison A. Curnutte, Tatjana Trcek

**Affiliations:** Homewood Campus, Department of Biology, Johns Hopkins University, 3400 N. Charles Street, Baltimore, MD 21218, USA; siran.tian@jhu.edu (S.T.); hcurnut1@jhu.edu (H.A.C.)

**Keywords:** RNA granules, p-bodies, stress granules, germ granules, RNA-RNA interactions, RNA self-assembly, RNA phase separation, RNA secondary structure

## Abstract

RNA granules are ubiquitous. Composed of RNA-binding proteins and RNAs, they provide functional compartmentalization within cells. They are inextricably linked with RNA biology and as such are often referred to as the hubs for post-transcriptional regulation. Much of the attention has been given to the proteins that form these condensates and thus many fundamental questions about the biology of RNA granules remain poorly understood: How and which RNAs enrich in RNA granules, how are transcripts regulated in them, and how do granule-enriched mRNAs shape the biology of a cell? In this review, we discuss the imaging, genetic, and biochemical data, which have revealed that some aspects of the RNA biology within granules are carried out by the RNA itself rather than the granule proteins. Interestingly, the RNA structure has emerged as an important feature in the post-transcriptional control of granule transcripts. This review is part of the Special Issue in the Frontiers in RNA structure in the journal Molecules.

## 1. Importance of RNA Granules

The self-assembly of RNA-binding proteins (RBPs) and RNAs by phase separation into predominantly spherical, non-membrane bound condensates called RNA granules is a phenomenon observed across species. These granules form in distinct cellular locations and compartmentalize biomolecules. This compartmentalization creates cellular asymmetries, may enhance biological reactions, and promote molecular interactions required for cell growth and development. Phase separation imparts liquid-like properties to these assemblies, allowing granules to rapidly condense and dissolve depending on the environment in which they form and enables the exchange of granule components with the granule environment [1].

Research of the past few decades uncovered a variety of RNA granules. Some, such as Processing bodies (P-bodies) and stress granules (Figure 1A,B) are wide spread, while others such as germ granules (Figure 1C–F) are highly specialized and form only in a subset of cells. Given that they enrich mRNAs, RNA granules have been mainly associated with post-transcriptional regulation. Increasing evidence suggests that these condensates appear at the center of diverse biological processes and are even associated with various human diseases (reviewed in [2]). For example, P-body homeostasis appears to regulate stem cell potency by alternating the chromatin structure of these cells [3]. In addition, P-bodies are involved in post-transcriptional regulation of the huntingtin protein in Huntington’s disease [4]. Stress granules have been found to promote cancer cell survival (reviewed in [5]) and may also regulate viral replication upon infection (reviewed in [6]). In *Drosophila melanogaster*, mutations in germ granule components induce sterility of the progeny (reviewed in [7]) whereas mutations of P granule components in *Caenorhabditis elegans* induce piRNA-initiated transgenerational silencing of genes involved in RNA interference [8]. Several features are conserved among diverse RNA granules, including their morphology, how they form, and what protein families enrich in them. Despite these similarities RNA granules have distinct biological roles. Studying different RNA granules therefore not only deepens our understanding of these condensates but also provides fundamental new insights into human disease.

## 2. Types of RNA Granules and Their Functions

### 2.1. P-Bodies

P-bodies have been first described in yeast cells in 2003 [9] and have since been intensely studied. Several observations indicated that these RNA granules could be primary sites of mRNA degradation. First, purification and imaging of P-bodies revealed that most proteins residing within them are involved in mRNA decay (i.e., XRN1, DCP1), mRNA surveillance (i.e., UPF1, SMG5), translation regulation (i.e., eIF4E, DDX6), and miRNA pathways (i.e., AGO1 and AGO2) (reviewed in [10,11]). Second, inhibiting mRNA decay by blocking XRN1, a highly conserved eukaryotic exoribonuclease responsible for the 5′-3′ degradation of cytoplasmic mRNAs [12], increases the size and the number of P-bodies, as well as the decay intermediates within P-bodies [9,13,14]. Third, P-bodies are not observed when deadenylation, the first step in mRNA decay pathway is blocked, but are restored after this blockage is removed [15].

P-bodies have long been seen as one of the cornerstones of RNA biology capable of organizing reactions fundamental to the physiology of a cell. However, a closer look challenges this view. Live cell imaging of single, fluorescently-labeled mRNAs revealed that in human epithelial (HeLa) cells mRNAs that enter P-bodies do not necessarily degrade in them [16]. Furthermore, a transcriptome-wide analysis revealed that 5′ truncated RNAs are mostly absent from purified P-bodies [17] and that kinetics of degradation and fates of different mRNAs within P-bodies vary such that some transcripts are degraded whereas others remain stably enriched in P-bodies [18,19]. In some instances, mRNAs that accumulate in P-bodies appear to be better protected from decay rather than being subjected to it [18]. Perhaps most surprising is the observation that most transcripts decay just as likely within granules as they do outside [16,17,20,21]. Thus, the P-body formation appears to be a consequence rather than a cause of RNA-mediated silencing [20]. Indeed, readily observable P-bodies do not form in budding yeast under normal conditions without a notable effect on mRNA turnover and instead appear when cells enter stationary growth or when they experience nutrient deprivation [13]. In addition, inhibition of mRNA decay induces P-body formation in vivo indicating that undegraded transcripts are necessary for their formation [13], further questioning the idea that P-bodies are required for mRNA turnover.

P-bodies could instead simply store transcripts and direct their flow through a translational cycle. For example, mRNAs appear to move between the polysome-depleted, P-body-enriched state and polysome-bound, cytoplasmic state [17,22,23,24,25,26]. This observation raises a possibility that P-bodies could facilitate translation re-initiation of repressed mRNAs by releasing them into the translation pool upon removal of cellular stress. However, this idea has similarly been challenged by demonstrating that decay rates of mRNAs, which accumulate in P-bodies were similar to decay rates of mRNAs that remain outside [21]. Therefore, the main function of P-bodies remains elusive and could instead be dependent on the cellular contexts and environmental conditions. Lastly, and perhaps speculatively, P-bodies could simply form as a manifestation of liquid-liquid phase separation without providing a particular function in regulating RNA biology.

### 2.2. Stress Granules

Stress granules form when messenger ribonucleoproteins (mRNPs) become stalled in translation initiation and can be induced by various cellular stresses such as heat shock, hypoxia, endoplasmic reticulum (ER) stress, arsenite poisoning, and viral infections [27,28,29,30,31]. They contain small ribosomal subunits (40S), translation initiation factors (i.e., eIF2, eIF3, eIF4A, eIF4G), and specific (marker) RBPs such as T-cell restricted intracellular antigen 1 (TIA-1), TIA-1-related protein (TIAR), and RasGAP SH3-domain binding protein 1 (G3BP1) and 2 (G3BP2) [32,33], which are largely depleted from P-bodies [17]. Unlike P-bodies, where over 70% of the proteome comprises RBPs, almost 50% of stress granule proteins are neither RBPs nor prion-like [34]. Stress granules tend to accumulate translationally-repressed mRNAs and disassemble quickly upon stress removal [35,36]. Due to their physical association with P-bodies (Figure 1A) and their potential to store translationally-repressed transcripts, stress granules have been hypothesized to function as triage sites that store mRNAs during cellular stress and direct them towards P-bodies for degradation or back into the cytoplasm for translation once the stress is relieved [33].

Purification and single molecule imaging techniques have also been applied to investigate the regulation of mRNAs in stress granules. These techniques revealed that while 185 mRNAs strongly accumulate in stress granules, most cellular transcripts enrich only at low levels (up to 30% of a given mRNA) [21,29,37]. Furthermore, after recovery from stress, those mRNAs that accumulated in stress granules became as efficiently translated as those that remained outside, indicating that accumulation of transcripts in stress granules does not destine them for a particular translational regulation [21]. In addition, the exchange of mRNAs between stress granules and P-bodies appears to be highly infrequent [21], indicating that stress granules might not serve as triage sites for all cellular transcripts. Perhaps most surprising is the fact that at least one mRNA, *ATF4* (an mRNA whose protein codes for a transcription factor that mounts a stress response in mammalian cells), translates abundantly and equally efficiently in stress granules as outside [38]. These data suggest that stress granules might not be always required for translational repression during stress. Indeed, U2OS cells lacking G3BP1 and 2, two redundant stress granule nucleators, do not form stress granules during sodium arsenite-induced oxidative stress but still exhibit stress-induced translational arrest [39]. What function these mysterious condensates perform in regulating RNA biology remains unclear.

### 2.3. Germ Granules

Germ granules form only in germ cells and appear inextricably linked with the germ cell biology. First described in the beetle *Calligrapha punctate* by Robert Hegner over 100 years ago [40], these condensates appear to regulate post-transcriptional processes to direct the germ cell fate across metazoans. Some germ granule components such as a DEAD-box RNA helicase Vasa (DDX4 in mammals and GLH-1 in *C. elegans*) and *nanos* (*nos*) mRNA are essential germ granule components and conserved across species (reviewed in [7,41]) while others such as Oskar, MEG-3/4, and Bucky ball, core granule proteins that nucleate germ granules in *Drosophila*, *C. elegans*, and *Danio rerio*, respectively are species-specific [42,43,44,45]. Germ granules are unique in that they associate with the same cell lineage throughout the development of an organism but can transform their morphology and function. For example, during early development, germ granules referred to as nuage envelop the nuclei within the germ cell lineage (Figure 1Ci,ii,Ei,ii) [46]. Nuage is thought to be involved in the biogenesis of small RNAs. It appears tightly associated with nuclear pores and both nuclear pore complex-like FG repeat components and mRNAs are found within these perinuclear germ granules [41,46,47,48]. During early embryogenesis, however, germ granules form in the germ plasm, a specialized cytoplasm that in *D. melanogaster* and *C. elegans* accumulates at the posterior pole of the early zygote (Figure 1Di,ii,F) (reviewed in [7,41]). These germ granules, also called polar granules (*Drosophila*) or P granules (*C. elegans*), enrich mRNAs whose protein products are required for the specification of the germ cell fate and thus temporally and spatially instruct the formation of the germ cell linage in these two organisms (reviewed in [7,41]).

#### 2.3.1. Germ Granules of the Early Embryo

Super-resolution microscopy revealed that in the early *Drosophila* embryo, the core germ granule proteins are homogeneously mixed within granules [49] and display liquid-like as well as gel-like properties [50]. The germ plasm and its germ granules (Figure 1Di,ii) are critically required for the establishment of germ cells as strong mutations of their core proteins inhibit granule formation and give rise to sterile progeny [51,52]. These granules are also polysome-bound [53] suggesting that germ granules could be sights of translational activity. Indeed, their purification revealed that they are composed of diverse RBPs involved in translational control, mRNA turnover, as well as RNA splicing [54,55]. Nearly 200 maternally deposited mRNAs such as *Cyclin B* (*CycB*), *nos*, *polar granule component* (*pgc*), and *germ-cell-less* (*gcl*) enriched in these granules [56], where they translate to control the formation (*gcl*), specification (*nos*), transcriptional silencing (*pgc*), and division (*CycB*) of the primordial germ cells (reviewed in [7,57]). While direct evidence for translational control in *Drosophila* germ granules is lacking, genetic and imaging data indicate that mRNAs that are granule-enriched predominantly translate within granules. For instance, *nos*, which codes for a translational regulator Nanos required for body patterning of the early embryo, as well as proliferation and migration of germ cells (reviewed in [7]), contains an evolutionary conserved translational control element (TCE) within its 3′UTR, which folds into a well-defined two stem RNA structure [58,59]. The TCE binds Smaug, a protein that prevents translation of unlocalized *nos* [60]. Upon localization into germ granules, Oskar, which is almost exclusively found in germ granules [50], prevents deadenylation and translational repression of *nos* imposed by Smaug [61]. A posterior-anterior gradient of Nanos protein thus forms, which instructs the formation of abdominal segments within the developing embryo [62,63]. In the absence of germ granules, the translational repression of *nos* is not relieved and the lack of Nanos causes embryonic lethality [62,63].

In contrast to polar granules of *Drosophila*, the protein components of P granules of the early *C. elegans* embryo are structured. Here, the gel-like shell formed by two intrinsically disordered proteins MEG-3 and its redundant partner MEG-4 envelopes the liquid-like core composed of PGL-1 and -3 proteins [64,65,66]. Purification of P granules identified 492 maternally-provided mRNAs as P granule-enriched, including *nos*-1 and 2 [67]. These two mRNAs are required for the specification of the germ cell fate in the worm [68]. However, unlike mRNAs enriched in *Drosophila* polar granules, mRNAs in P granules in *C. elegans* are translationally repressed and only translate once they become released from P granules [67]. In addition, P granules are dispensable for the establishment of germ cells in *C. elegans* and their removal results in only approximately 30% reduction in the fecundity of the offspring [69].

#### 2.3.2. Nuage

Nuage is involved with the production of small, piwi protein-interacting RNAs (piRNAs) that, in *Drosophila,* mediate transposon silencing (reviewed in [70]) and, in *C. elegans*, the self/non-self recognition of germline mRNAs (reviewed in [71]). During self/non-self recognition (a mechanism that allows the worm to distinguish foreign RNAs from its own), the nuage harbors transcripts required for RNA-mediated interference. In the absence of nuage, these transcripts become dispersed through the germ cell cytoplasm where they are recognized by the piRNA machinery and degraded [8]. In addition, small 22G RNAs that arise from this piRNA-mediated degradation accumulate and feed-back into the nuclei to further suppress transcription of respective mRNAs, a phenomenon that is trans-generationally inherited [8]. Therefore, nuage appears to act as an mRNA export center that sequesters mRNAs susceptible to degradation by the piRNA machinery and protects them from piRNA-initiated transgenerational silencing [8].

In addition, different nuage functions appear to be segregated to different granules. For instance, in *C. elegans*, P granules, Z granules, and mutator foci associate into PZM assemblages at the nuclei of germ cells. These tri-condensate assemblages are believed to be involved in RNAi inheritance, with localized functions of mRNA surveillance, RNA marking for memory storage, and transposon surveillance in P granules, Z granules, and mutator foci, respectively [72].

Despite the different functions of P-bodies, stress granules and germ granules in post-transcriptional regulation, fundamental features are nevertheless shared among these RNA granules. Perhaps most surprising is the fact that many granule properties are instructed by the transcripts themselves rather than the granule proteins, as discussed below.

## 3. Formation of RNA Granules: The Role of the RNA

RNA granules form when the concentration of a protein which nucleates them reaches a critical concentration that triggers condensation [1]. Intrinsic properties of granule proteins including RNA-binding domains, intrinsically disordered regions, amino acid composition, and their capacity to engage in multiple interactions with neighboring molecules (valency) govern the physical properties of phase separation [73,74]. In addition, environmental factors such as molecular crowding, temperature, pH, and osmolarity also contribute by modulating the critical concentration required for condensation [75]. Notably, RNAs are active regulators rather than passive passengers in granule formation. In vitro, a small amount of RNA initially promotes protein droplet formation and afterwards begins dissolving the droplets once the concentration of the RNA reaches a threshold [76,77]. RNAs change physical properties of condensates (i.e., viscosity), augment granule assembly in vitro and in vivo [13,44,67,76,77,78,79,80,81,82,83,84], determine the three dimensional structure of granules in an RNA concentration-dependent manner [49,85], and can even instruct the morphology of granules by their ability to engage in intermolecular RNA-RNA interactions [86].

### 3.1. The Role of Long and Ribosome-Free RNAs

Long and translationally-repressed (exposed) transcripts tend to enrich in RNA granules (Table 1) [17,29,37,67,87]. These RNAs can have more conformational states [88] and could also engage in more protein-RNA interactions, promiscuous or sequence-specific intermolecular RNA-RNA base-pairing, or non-Watson-Crick interactions (Figure 2) [79,83,89,90,91,92]. Electrostatic interactions between proteins and RNA could also enhance granules formation (Figure 2). These interactions have been widely studied in viral nucleocapsid proteins, where viral proteins act as RNA chaperones that mask negative charges of nucleic acids. Recent fluorescence resonance energy transfer (FRET) experiments demonstrated that RNA chaperones act as macromolecular counterions that facilitate packing of negatively charged nucleic acids [93]. Therefore, electrostatic interactions may play a similar role in granule assembly. For example, G3BP1 RBP, a core stress granule protein [29,32,37] has three intrinsically disorder regions termed IDR1, IDR2, and IDR3. IDR1 is highly negative charged, IDR2 is slightly positively charged while IDR3 is an arginine-glycine (RG)-rich RNA-binding domain and net positively charged [94]. By using genetic mutants, Yang et al. found that IDR1 is inhibitory for stress granule formation while IDR3 augments it [94]. Since the electrostatic charges of RNAs and proteins could be shielded within tightly packed RNA condensates (see below), it is possible that the granule-associated proteins and RNAs have different conformations within and outside the granules. How RNA and protein conformations affect granule assembly and whether such interactions are mainly driven by electrostatic forces applicable to diverse granules requires further experimentation.

### 3.2. The Role of RNA Structure

Interestingly, the RNA structure is emerging as one of the principal regulators of phase separation (Figure 2). Highly structured RNAs tend to have more interactions with RBPs than less structured transcripts and can rearrange the composition of protein aggregates [95]. The secondary structure determines whether an mRNA will be recruited into the condensate and whether it will promote the assembly of distinct RNA granules within a cell [79]. In addition, secondary structures expose or mask the sequences for intermolecular RNA-RNA interactions [79] further influencing condensation. RNA structures can change upon nucleotide modifications, such as *N*^6^-methyladenosine (m^6^A), the most common mRNA modification [96]. In mouse embryonic stem cells, m^6^A impacts the RNA structure by inducing the transition from paired to unpaired RNA strands [97] and exposes RNA sequences to protein binding [98]. In addition, in vitro m^6^A enhances phase separation of proteins that bind m^6^A [78] indicating that RNA modifications or secondary structures these modifications induce contribute to the specificity of phase separation. Therefore, it is not surprising that mammalian stress granules enrich m^6^A-modified mRNAs [99,100] and that the levels of m^6^A modification in stress granules are approximately 50% higher than outside of granules [78]. Similarly, *Drosophila* of many polar granule-enriched mRNAs are similarly m^6^A modified [101]. Thus, it is possible that modified mRNAs may act as scaffolds to help granule proteins condense. Indeed, the YTH domain-containing family (YTHDF) proteins that bind m^6^A modifications enrich in mammalian stress granules and promote their formation [99]. The diverse RNA-protein and RNA-RNA interactions driven by exposed RNA sequences, structures, and modifications could create a highly interconnected molecular network that would in turn increase the stability of granules (Figure 2). Indeed, in vitro experiments and live imaging of *Drosophila* germ granules demonstrated that RNAs are more stably associated within granules than the granule proteins themselves [80,85]. RNAs could therefore constitute the stable component of granules capable of tuning the phase separation in cells.

## 4. Enrichment of RNAs to Granules: RNA Could Self-Recruit

### 4.1. Mechanisms of Enrichment

Enrichment of mRNAs to most RNA granules is passive. For instance, live imaging of *nos* in *Drosophila* oocytes demonstrated that this mRNA localizes to germ granules by a diffusion-entrapment mechanism and does not involve mRNA transport via molecular motors [102]. Characterization of purified RNA condensates also revealed that non-translating mRNAs and some long non-coding (lncRNAs) make up the transcriptome of stress granules and P-bodies, as well as germ granules in *C. elegans* (Table 1) [17,18,29,37,67,103,104]. While mRNAs in *Drosophila* polar granules appear to translate, a genetic analysis revealed that mRNAs recruited to *Drosophila* germ granules also tend to be translationally-repressed [58,59,87,105]. These data suggest that translational repression is a major determinant in the recruitment of RNAs to diverse RNA granules (Table 1).

Recently, single molecule imaging in U2OS cells provided direct evidence for these observations. Moon et al., demonstrated that translating mRNAs cannot become stably recruited to stress granules until the ribosomes have run off of the mRNA and translation has ceased [106]. These techniques also revealed that within P-bodies in U2OS cells, miRNAs are bound to their targets but when these targets were modified such that they no longer base-paired with repressive miRNAs, mRNA localization and enrichment to P-bodies decreased significantly [104]. Thus, ribosome dissociation from an mRNA is a pre-requisite for stable recruitment of most RNAs to P-bodies and stress granules [14,28,29,106,107]. However, not all translationally repressed mRNAs are recruited to stress granules [21,29] and at least one mRNA, *ATF4*, is stably associated with stress granules despite being translated within them [38]. This finding suggests a surprising new aspect of mRNA regulation in this mysterious RNA condensate. Similarly, translational repression correlates with RNA enrichment to germ granules in *Drosophila*, but it is not a requirement for enrichment. For instance, *CycB* is recruited to polar granules even though it translates outside of granules [108]. In addition, sucrose density gradient experiments revealed that unlocalized *nanos* remains associated with polysomes even though it is translationally-repressed [109] indicating that ribosome-free mRNA is not required for enrichment of *nos* to *Drosophila* polar granules. In support of this observation, deletion of the *nos* TCE abolishes the spatial control of translation of *nos* without affecting its ability to enrich in polar granules (Table 1) [59,110,111]. Finally, not all long mRNAs or long non-coding RNAs (lncRNAs) enrich in granules. Some lncRNAs interact with P-bodies only transiently and are largely depleted from the P-body transcriptome [17,104]. Thus, translational repression may be one of the mechanisms, but not a sole requirement for recruitment of RNAs to RNA granules.

### 4.2. Efficiency of Enrichment

RNA enrichment in condensates is an inefficient process. Up to four percent of a particular mRNA is enriched in *Drosophila* germ granules [49,110], which comprise about 0.01% of the embryo volume [49], while the rest of the mRNA remains dispersed through the embryo. Similarly, only a small fraction of mRNAs accumulates in P-bodies and stress granules, with the bulk remaining dispersed in the cytoplasm [21,29,37]. Additionally, the transcriptome of a particular RNA granule changes depending on the cell type or the environmental condition. For example, stress granules recruit translationally-repressed transcripts, however, the transcriptome accumulated in stress granules, which form during ER stress is somewhat different from the one that accumulates in stress granules during arsenite poisoning or heat stress [37]. In addition, during ER stress, almost 99% of mRNAs become strongly translationally repressed and yet only about 20% of these enrich in stress granules [37]. Finally, P-bodies that form in non-stressed conditions and stress granules recruit different sets of translationally-repressed mRNAs: P-body transcriptome is composed of mRNAs entirely devoid of ribosomal components while mRNAs that enrich in stress granules can still be bound by 40S ribosomal subunits (reviewed in [11]). Interestingly, during stress, P-body transcriptome shifts to resemble the transcriptome of stress granules [112], suggesting that environmental conditions can influence the mRNA composition of RNA granules. These data further support the idea that mechanisms other than translational regulation could contribute to the efficiency and the specificity of RNA recruitment to granules [8,37].

### 4.3. Specificity of Enrichment

Despite seeming permissiveness, mRNA enrichment to RNA granules is nevertheless highly selective. For example, in *Drosophila*, the 3′UTR is the major determinant of mRNA enrichment to polar granules [87,105,111], suggesting that these RNA segments carry particular sequences that mediate interaction of mRNAs with polar granules. Indeed, of the nearly 6000 genes expressed in the early *Drosophila* embryo [113] only up to 200 enrich at the posterior pole [56]. Similarly in *C. elegans*, of the 8890 genes expressed in the early embryo [114], fewer than 500 distinct mRNAs enrich in P granules [68]. In human U2OS cells, only 185 mRNAs enrich more than 50% efficiently in stress granules [29]. Interestingly, polar granule, P granule, and stress granules are composed of RBPs capable of interacting indiscriminately with the majority of cellular transcripts [50,54,115] and therefore, individually, these RBPs cannot provide the specificity for mRNA enrichment observed in vivo. In *Drosophila*, in vivo crosslinking and immunoprecipitation (CLIP) assays combined with in vitro RNA-binding analysis identified the germ granule nucleator Oskar as a non-canonical RNA binding protein capable of interacting with a few germ granule mRNAs such as *nos*, *pgc*, and *gcl* with a low affinity and weak sequence specificity [116,117]. Similarly, the P granule nucleator MEG-3 in *C. elegans* is also an RBP that interacts with mRNAs without sequence specificity and “prefers” longer transcripts with low ribosome occupancy [68]. To ensure selective mRNA recruitment, the RNA binding could be combinatorial involving multiple RBPs or could instead involve the RNA itself. Indeed, data suggests that mRNA enrichment to *Drosophila* germ granules and mammalian stress granules could in part occur through mRNA self-recruitment, where mRNAs anchored within granules could recruit additional mRNAs independently of granule proteins [92,118]. The mechanism for self-recruitment is unknown. However, intermolecular RNA-RNA interactions mediated by RBPs or by direct Watson-Crick base pairing could provide one mechanism. Longer, non-translating mRNAs with exposed sequences may provide more valency for RNA-protein and RNA-RNA interactions and increase the likelihood of self-recruitment. In support of this hypothesis, RNA helicases that unwind secondary structures and base-paired RNA stretches modulate RNA condensation in vitro and in vivo and affect the transcriptome of stress granules in U2OS cells in vivo [92]. RNA self-recruitment could also increase the efficiency of enrichment by eliminating the competition for entrapment by granule proteins. Thus, via self-recruitment, mRNAs could directly influence the transcriptome of RNA granules and thus the function of the granule.

### 4.4. Conserved RNA Properties of Diverse RNA Granules

Characteristics of RNAs enriched in P-bodies, stress granules, and germ granules (Table 1) pertain to other RNA granules as well. For example, bacterial RNP-bodies (BR-bodies) “prefer” long and non-translating mRNAs [119]. Similarly, fragile X mental-retardation protein FMRP, an mRNA-binding protein that forms granules in mammalian neurons [120,121], also selects for longer, translationally-repressed mRNAs [122]. In addition, mRNAs that encode membrane proteins use AU-rich elements located in their 3′UTRs to enrich TIS-granules (membraneless organelle associated with the endoplasmatic reticulum) [123], similar to the mRNA enrichment mechanism found in *Drosophila* polar granules [87,105,111]. Thus, aside from their morphology and biophysical properties, RNA enrichment principles seem conserved among diverse RNA granules.

## 5. RNA Organization in RNA Granules: RNAs Self-Organize

### 5.1. Role of trans RNA-RNA Interactions

RNAs can spatially-organize independently of RBPs using intermolecular RNA-RNA interactions. *Drosophila bicoid* (*bcd*) and *oskar* (*osk*) mRNAs, the RNA genome of the human immunodeficiency virus (HIV) base-pair with each other in vitro and in vivo in RNA sequence-dependent manner [91,126,127,128,129]. Organization of *bcd* and *osk* into multi-mRNP transport particles drives their enrichment to the opposite poles of the embryo and helps establish the body plan of the developing organism. However, genetic mutations, which drive inappropriate RNA-RNA interactions cause neurodegeneration. For instance, over-expanded CUG repeats in the 3′ untranslated region (UTR) of the myotonic dystrophy protein kinase (DMPK) gene are thought to engage in multivalent base-pairing, thus driving the formation of nuclear RNA inclusions [91]. These inclusions are the causative agent of Myotonic Dystrophy 1 (DM1), driving progressive muscle weakening and premature death [130,131].

RNA assemblies can also form *via* RNA base stacking, which can happen both with and without the Watson-Crick base pairing [132]. Within the adjacent base-pairings, the bases may stack on each other either within the same strand or the opposite strand, possibly helping to stabilize the RNA dimerization as demonstrated for HIV dimers, bacterial Host factor-I bound RNAs as well as *bicoid* mRNA dimers [132,133,134]. The G-quadruplex, on the other hand, is based on the interactions among the aromatic rings of the guanines via Hoogsteen-type hydrogen bonding. Depending on the orientation of each guanine base, guanine base-stacking in the G-quadplex may alter the RNA geometry, RNA thermodynamics and translation [135,136] and also facilitate RNA phase separation both in vitro and in vivo [137,138,139].

Increasing evidence suggests that intermolecular RNA-RNA interactions also play a major role in the formation of RNA granules. Non Watson-Crick, non-specific intermolecular RNA:RNA, as well as sequence-specific intermolecular RNA-RNA interactions can promote homotypic and heterotypic RNA assemblies [79,89,90,91,92]. For instance, in vitro polyA and polyU RNAs heterotypically co-assemble while polyC and polyU de-mix into individual homotypic droplets that touch each other with a clearly defined border [92]. Thus, RNAs that are more likely to base pair with each other are also more likely recruited to the same condensate where they co-assemble [91,92]. In vitro, these base-paired RNAs adopt a rigid, highly compacted and poorly mobile architecture within the aggregate [91].

In vivo, mRNAs in *Drosophila* germ granules also self-assemble into homotypic clusters [85], which contain multiple mRNPs derived from the same gene (Figure 3A–D) [49,125]. Homotypic clusters from different genes co-exist and de-mix from each other within the same granule (Figure 3C,D) with more abundant clusters (those that contain multiple transcripts per cluster) residing in the center of the granule and less abundant ones residing at the granule periphery (Figure 3A,B) [49,85]. The concentration of mRNAs within homotypic assemblies is on average 4000-fold higher compared to the outside [85] indicating that mRNAs within clusters are highly compacted. Transcripts that accumulate in stress granules are also highly compacted and form a much smaller footprint within granules than in the cytoplasm [27,140]. These data suggest that localization into granules likely induces a conformational change in the RNA structure that could promote intermolecular RNA-RNA interactions allowing compaction. However, unlike polyU, A, G, and C RNAs and mRNAs with expanded repeats, which co-assemble via sequence-specific base-pairing (Figure 3D) [91,92], homotypic clusters in *Drosophila* germ granules self-assemble in an RNA sequence-independent manner without detectable sequence-specific intermolecular RNA-RNA interactions (Figure 4) [85]. These homotypic mRNA assemblies display a much reduced mobility compared to the proteins that form germ granules [85] raising a possibility that instead multiple sequence-independent, non-Watson-Crick, as well as RNA-protein interactions could stabilize mRNAs within a highly compacted cluster environment. Interestingly, super resolution microscopy revealed that despite an increased compaction within homotypic clusters, multiple non-specific RNA-RNA interactions are insufficient to spatially-constrict mRNAs to the same degree as a single, sequence-driven *trans* RNA-RNA interaction [85].

### 5.2. Homotypic mRNA Self-Assembly

It is unclear how mRNAs could self-assemble homotypically. In *Drosophila*, genetic experiments using chimeric reporters failed to identify mappable mRNP regions required to generate the specificity of homotypic assembly. Instead, small perturbations in the primary RNA sequence allowed chimeric transcripts to de-mix from the endogenous counterparts (Figure 4), arguing that a global property of the mRNP rather than a particular RNA feature alone or in combination likely provides the specificity for homotypic self-assembly in vivo [85]. In this model, mRNPs with different miscible properties specified by associated proteins, RNA modifications and structures could control de-mixing. Here, an mRNP could “read” the sum of all of its characteristics to discriminate between transcripts rather than rely on one single determinant such as an RNA fold. An implication of this model is that transcripts, which would share their primary sequences but became distinctly modified, structured or protein-bound, would also sort into distinct clusters. An average length of a *Drosophila* transcript is 3058 bases (estimate from BioNumbers [141]), allowing an mRNA to adopt a variety of secondary structures, carry multiple RNA modifications, and become bound by diverse proteins, all of which could influence the miscibility of an mRNA. These features could also change in response to cellular stimuli. For example, translating ribosomes could uncouple paired RNA duplexes. This is feasible given that transcripts that enrich in *Drosophila* germ granules translate once they become localized [87,105]. While mRNA self-assembly is not driven by granule proteins, these proteins are nevertheless critical for clustering as they provide a platform for the mRNAs to self-recognize. By entrapping transcripts and highly concentrating them, the granule proteins could act as crowding agents that increase the probability of transcripts to self-assemble and alter the properties of enriched mRNA [79]. For instance, recruitment of nucleic acids into phase separated granules melts double-stranded duplexes and therefore changes their structural properties [142], a process that could enable assembly of distinct RNA droplets within the same cell [79]. Thus, mRNA self-assembly could be a consequence of modified mRNA properties induced by granule proteins, allowing the mRNAs to organize specifically in granules and not outside.

### 5.3. Role of mRNA Self-Assemblies

mRNA assemblies have been observed in diverse RNA condensates, including germ granules in *Drosophila* [49,125] and zebrafish [143,144], P-bodies in *Drosophila* oocytes [145], in mammalian stress granules [29], and P-bodies [104]. Thus, homotypic assembly is an inherent property of mRNAs, revealed when they become sufficiently crowded, such as upon their enrichment in RNA condensates. Assembly could be functionally important. Just as assemblies locally increase mRNA concentration [85], they may also increase the concentration of important RNA regulators and thereby alter the post-transcriptional dynamics of granule mRNAs. For example, clustering could release mRNAs from translational repression or induce it, and even augment their translational efficiency by retaining ribosomes or translational regulators within assemblies to increase the likelihood of additional rounds of translation. Many *Drosophila* germ granule-enriched mRNAs are subjected to rapid degradation outside of germ granules and stabilized inside granules [113,146] and similar principles described above could also apply to increase the stability of granule-enriched transcripts. Given high compaction within granules that could promote *trans* RNA–RNA interactions (Figure 2) [27,85,140], mRNA assemblies could also act as structural components that would help form and maintain granules [13,80,81,89,147,148].

It is unclear why some RNA assemblies, such as those that form in RNA granules appear benign or perhaps beneficial while others, such as those manifested in DM1, cause disease. The mRNA assemblies observed in RNA granules form within a proteinaceous granule environment, which could prevent possible harmful RNA aggregation observed for toxic RNA inclusions in DM1. Indeed, RNA helicases are core components of diverse RNA granules, which could dissolve potentially harmful RNA-RNA interactions that may form within highly compacted RNA assemblies. Alternatively, the location within a cell where mRNAs assemble could also be relevant. For instance, if the granule RNAs self-assembled in the nucleus rather than in the cytoplasm, as is the case with RNA inclusions in DM1, perhaps they too could become pathogenic. Future experimentation is needed to elucidate the role of RNA self-assemblies in normal cell growth as well as in disease.

## 6. Outlook

Enrichment of RNAs in RNA granules provides compartmentalization of post-transcriptional processes within a cell. The role of many RNA granules in post-transcriptional regulation remains unclear and requires further investigation. Surprisingly, RNA itself regulates many aspects of its own biology within RNA granules, including its recruitment to and organization within granules as well as stabilization of granules. From this perspective, RNA granules remind us of the primordial world in which the RNA was the “working” molecule that organized fundamental reactions. Thus, intriguingly, RNA granules could be seen as the remnants of the early world, in which the RNA takes center stage and carries out important functions, while the granule proteins provide the supporting role, enabling the RNAs to do their work. As for the past few decades, these mysterious condensates continue to surprise us and undoubtedly, future discoveries will reveal fascinating and fundamental new roles of granule RNAs in post-transcriptional regulation.

## Figures and Tables

**Figure 1 molecules-25-03130-f001:**
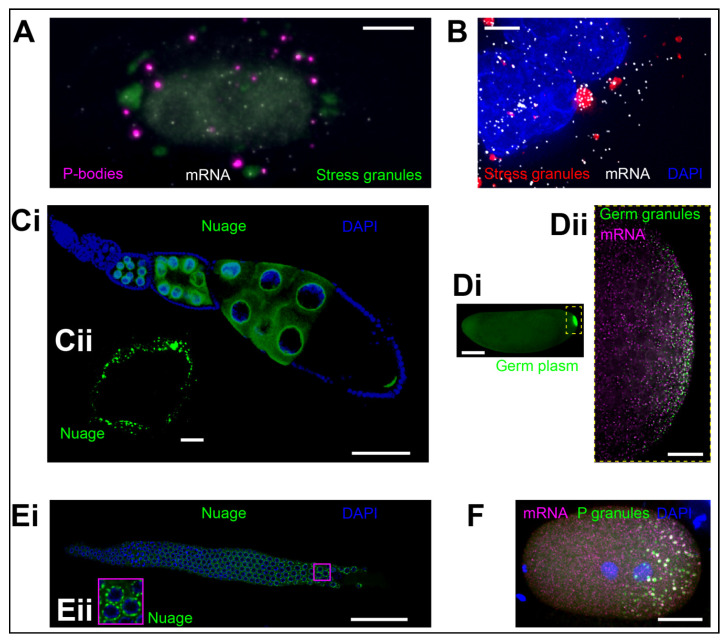
(**A**) Processing bodies (P-bodies) (magenta, marked by DDX6 fused with TagRFP) interacting with stress granules (green, marked by G3BP1 fused with GFP) in U2OS cells during stress. mRNAs (a transgenic mRNA genetically-tagged with MS2:MCP-Halo) are shown as white dots. Image: Courtesy of Jeff A. Chao (FMI). (**B**) Stress granules (red; marked by G3BP1 fused with GFP) accumulate AHNAK mRNA (white, hybridized with smFISH probes) in U2OS cells during stress. Nuclei are labeled with a DAPI stain. Image: Courtesy of Stephanie Moon (University of Michigan) and Roy R. Parker (University of Colorado, Boulder). (**Ci**,**ii**) Nuage (green, marked by Vasa fused with GFP) abutting the nurse cell nuclei (blue, marked by DAPI) in *Drosophila* oocytes. (**ii**) Shows a close-up of a single nucleus from a different oocyte from the one shown in (**i**). (**Di**,**ii**) Germ granules (polar granules) (**ii**) form within germ plasm (**i**) at the posterior pole of the early *Drosophila* embryo (green; marked by Vasa fused with GFP). mRNAs such as *gcl* (magenta, hybridized with smFISH probes) enrich in these granules. (**Ei**,**ii**) Nuage ((**ii**) magenta square: Close-up of three nuclei) (green, marked anti-CSR-1 antibody) abutting the *C. elegans* germ cell nuclei (blue, marked by DAPI). Image: Courtesy of Jessica Kirshner and John K. Kim (Johns Hopkins University). (**F**) mRNAs in P granules are enriched in the *C. elegans* one cell zygote. P granules enrich polyA RNA (green) and *nos*-2 mRNA (magenta), both marked with smFISH probes. DNA is stained with DAPI (blue). Image: Courtesy of Madeline Cassani, Andrew Folkmann, and Geraldine Seydoux (John Hopkins School of Medicine). Scale bar in (**A**,**B**,**Cii**) is 5 µm, in (**Dii**,**F**) is 10 µm, in (**Ci**,**Ei**) is 50 µm, and in (**Di**) is 100 µm.

**Figure 2 molecules-25-03130-f002:**
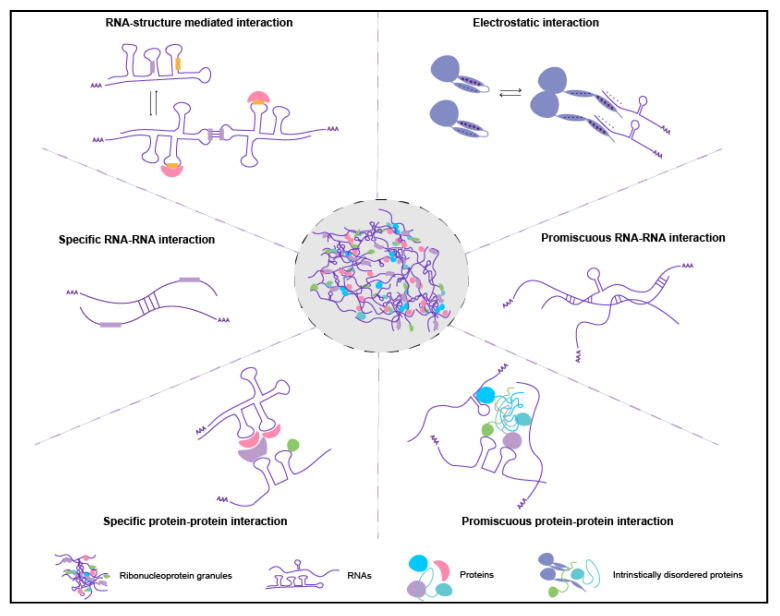
Proposed intermolecular interactions among RNAs and proteins within RNA granules. RNA secondary structures can mask/expose the specific RNA sequences or binding motifs to neighboring RNAs or RBPs. The electrostatic interactions may facilitate RNA-protein interactions in crowded environments, and the proteins can change the conformations when the RNA binds. In addition, intermolecular RNA-RNA and protein-protein interactions can be specific or promiscuous. Intermolecular RNA-RNA and RNA-protein interactions depicted in this model were mostly studied on mRNAs, however similar principles may apply to other RNAs enriched in RNA granules such as piRNAs, microRNAs, or lncRNAs.

**Figure 3 molecules-25-03130-f003:**
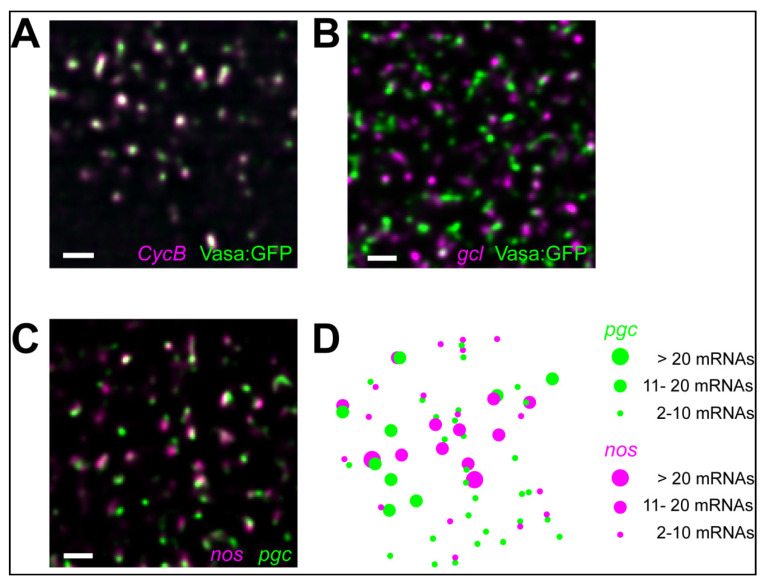
(**A**,**B**) *CycB* and *gcl* mRNA clusters (magenta; hybridized with smFISH probes) occupy distinct positions within *Drosophila* germ granules (green, labeled by Vasa fused with GFP). (**C**,**D**) Homotypic clusters contain multiple *nos* or *pgc* mRNAs (magenta and green, respectively hybridized with smFISH probes), do not have a defined stoichiometry and de-mix from each other within the same granule [85]. Scale bar in all is 1 µm.

**Figure 4 molecules-25-03130-f004:**
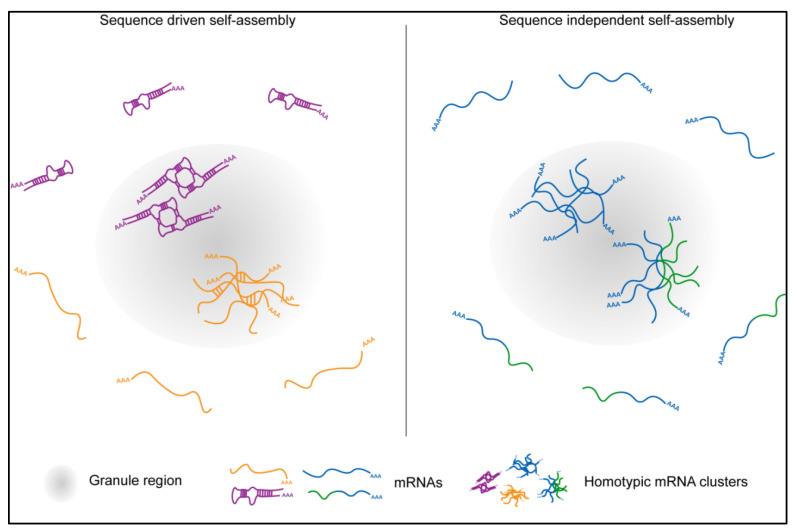
Proposed models showing the mechanisms of mRNA self-organization in RNA granules using sequence-dependent or sequence-independent assembly. The later mechanism was demonstrated for homotypic mRNA clusters in *Drosophila* germ granules. Here, mRNAs distinguish between endogenous mRNAs and its derivatives and de-mix to form distinct homotypic clusters [85].

**Table 1 molecules-25-03130-t001:** Characteristics of RNAs enriched in RNA granules.

RNA Granule	Enriched RNAs (Examples)	Depleted RNAs (Examples)	Prefer Long mRNAs?	Prefer Translationally Repressed mRNAs?
P-bodies	Translationally repressed mRNAs [17]	18S and 28S rRNAs [17]	N/A	Yes
Stress granules	mRNAs with longer coding regions and 3′UTRs [29,37] lncRNAs [29] 18S rRNA [124]	Membrane-associated mRNAs [29] *gapdh* [29] 28S rRNA [124]	Yes	Yes
P granules (*C. elegans*)	Long mRNAs with low ribosome occupancy [67] Germ-cell specific mRNAs [67]	N/A	Yes	Yes
Polar granules (*D. melanogaster*)	Germ-cell specific mRNAs [49,87,125]	*oskar* [49,125]	N/A	Yes, but with exceptions [58,59,87,105,108,109,110,111]

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
