# Peer review of "RNA Granules: A View from the RNA Perspective"

_molecules, 2020, doi:10.3390/molecules25143130_

Round 1

Reviewer 1 Report

In RNA Granules: A view from the RNA perspective submitted to molecules by Tian et al., (2020), the authors summarize recent findings of the RNA aspect of RNP granules suggesting RNA molecules, rather than being passengers, plays an active role in RNP granule assembly and organization. The authors divided the role of RNA in RNP assemblies into four sections. (1) How RNP assemblies affect the fate of resident RNAs. (2) The type of interactions RNAs can form that enable granule formation. (3) The types of RNAs that enrich in granules. (4) And how RNA properties determine RNA organization within granules. 

The authors also provided a brief introduction into three types of granules: P-bodies, stress granules, and germ granules.

           Although this is a much-needed review, several issues should be addressed before I recommend acceptance to MDPI molecules. First and foremost, there are a few scientific inaccuracies or clarifications I think the authors should address. Secondly, the manuscript has organization and writing issues, which makes some sections difficult to understand. Finally, the manuscript lacks a broader scope that would help non-granule readers realize the importance of RNP granules. Described below are specific comments on these issues and recommendations on how to address them.    

Scientific inaccuracies and/or my misunderstanding

1) In section 3, the authors make a point that translational repression may not be the only mechanism for enrichment in granules, which I agree, but from reading the review, I think the authors fail to point out that the current dogma in P-bodies and stress granules field is ribosome dissociation from RNA is a pre-requisite for stable RNA recruitment to granules and granule formation (Cougot et al., (2004), Andrei (2005) Teixeria et al., (2005), Kedersha et al., (2000), Khong and Parker (2018), Moon et al., (2018)). After ribosomes are released, other factors may come into play that determine what ribosome-free mRNAs accumulate in RNA granules. I recommend the authors state ribosome dissociation as a pre-requisite for RNA recruitment to granules as a general principle.

2) Moreover, the statement on lines 320-322 is inaccurate: “The P-body transcriptome attracts those mRNAs that appear almost entirely devoid of ribosomes while stress granules appear to “prefer” mRNAs that remain associated with a few ribosomes” with no reference provided that stress granules internalize RNAs with ribosomes associated. The authors might suggest Mateju et al., (2020) bioRxiv. See the next point.

3) The authors claim that translation might be happening inside stress granules (Mateju 2020 bioRxiv)) on lines 120-122. I do want to note the authors should be careful about using this as evidence of translation inside stress granules because overwhelming data (comment 1) suggest non-translating RNAs are internalized in granules and help facilitate granule formation and translating RNAs do not get internalized into granules. Secondly, 28S rRNAs are depleted from stress granules by ISH-EM indicating translation is not happening inside granules (Souquere et al., (2009)). Third, the preprint paper is not particularly convincing because the imaging resolution may not be sufficient for noting RNAs is inside or outside stress granules. Moreover, RNAs that are transiently interacting with stress granules (docking) can be found on the surface of stress granules (Moon et al., (2019), Moon et al., (2020)) which might explain why Mateju et al., (2020) bioRxiv see ATF4 translation “inside” stress granule. Regardless, if the authors do want to make this point, I think they should mention that there is a large body of work that suggests otherwise, and that this may be only unique to one mRNA, ATF4.  

4) The authors wrote on lines 319-320: “P-bodies and stress granules recruit distinct sets of translationally-repressed mRNAs.” While this is true when comparing non-stress P-bodies to stress granules, the transcriptomes are quite similar when comparing P-bodies under stress to stress granules (Matheny et al., 2019 Mol Cell Biol). Again, I recommend the authors note this, which is another evidence that translation plays a pre-requisite role in determining RNA localization to P-bodies and stress granules.

5) On lines 315-317, the authors wrote: “the transcriptome accumulated in stress granules, which form during ER stress is different from the one that accumulates in stress granules during arsenite poisoning or heat stress.” I want to note that this difference is minute. The correlation between these datasets in Namkoong et al., (2018) article is 0.67 and 0.79, so I recommend the authors indicate this because it may mislead readers that the transcriptomes are quite different.

6) The authors summarized in section 4.1 how RNAs can self-organize using sequence-specific and nonspecific interactions. I agree with most of the in vitro and in vivo data presented that supports RNAs can self-organize, however concerning homotypic clusters in Drosophila germ granules, I do not get the logic that it can be used as evidence for RNA self-organization. My understanding is the mechanism of how homotypic clusters form within germ granules (Trcek et al., 2020) was unresolved, and because it was unresolved, the authors speculate nonspecific RNA-RNA interactions might be driving this process but did not actually prove this is the mechanism. I would like the authors to lay out the logic for the reader that this is an evidence.  

7) On lines 263-264: The authors wrote, “modified mRNAs may act as scaffolds to help granule proteins condense,” citing Ries et al., (2019). Ries et al., (2019) only showed m6A plays a role in targeting specific modified RNAs to granules and do not provide evidence that they are important for stress granule formation in cells. The authors might want to cite Fu and Zhuang (2020) because they showed YTHDF interaction with m6A is important for facilitating stress granule formation in cells. Also, Anders et al., (2018) is another article the authors should mention because they are the first article to show that m6A modification may recruit modified RNAs to stress granules.

8) Since this article is about RNA’s role in granules, I would recommend mentioning/citing this article: RPS28B mRNA Acts as a Scaffold Promoting Cis-Translational Interaction of Proteins Driving P-body Assembly by Fernandes and Buchan (2020).

Organization and writing issues

1) The authors provide a general overview of P-bodies, stress granules, and germ granules. However, sprinkled throughout the text, the authors mention other types of granules that support different aspects of how RNAs plays a role in facilitating RNP assemblies. For example, nuclear RNA inclusions in muscular dystrophy or granules found in Ashbya gossypi. Moreover, the authors also cite many other granules in references like BR-bodies, neuronal granules, and TIS-granules. This can get quite confusing because, as a reader I am not sure if the RNA properties that play a role in RNP assemblies are general principles or specific to some RNP granules. A table consisting of different types of granules, what RNAs are enriched in these granules, and what types of interactions within these granules that’s been documented/speculated may be very helpful for readers. For example, is RNA length specific to only stress granules or is this a feature found in many other bodies. What about ribosome free RNAs? How about nonspecific or specific RNA-RNA interactions? RNA-modifications?

2) I find the logic hard to follow because of the way it is written. The first couple of sentences in each paragraph do not always give an indication of what the paragraph is about. Here are three examples.

(1) For example, on line 275 at the start of a new paragraph: the authors wrote: “enrichment of mRNAs to most RNA granules is passive.” But then went on to describe how it is not passive for most of the pargraph: B-actin mRNPs localization, zipcodes, RNA-BPs. Maybe the first sentence should be about targeting of mRNAs to granules is multifaceted.

            (2) Similar problems are found in other paragraphs, for example, line 206: “RNA granules form when the concentration of a protein which nucleates them reaches a critical 206 concentration that triggers condensation [1]. But then the rest of the paragraph is about RNA’s role (not protein) in facilitating granule formation. 

I would separate into two-three paragraphs, one with protein and then the second one with RNA. And perhaps last one about G3BP.

           (3) And line 151: “Super-resolution microscopy revealed that in the early Drosophila embryo, the core germ granules proteins are homogenously mixed within granules and display liquid-like…” but the bulk of the paragraph is spent on describing germ granules can be sites of translational activity. I think the paragraph should be led with Germ granules may be sites of translational activity.  

           This is a stylistic opinion, but I think it will be easier for readers to comprehend the review if the first couple of sentences provide a general indication of what the paragraph is about.

3) There are several improvements I would recommend for Figure 1. Figure 1A: stress granules, P-bodies, and mRNAs are quite small. I recommend zooming in on or two cells, so the smFISH spots are more noticeable, and the granules are bigger. Similarly, in Figure 1Dii and F, the mRNA and P-granules are also quite small, and I would recommend zooming in so readers can see individual RNAs. In Figure Cii, the close-up box for the corresponding Ci image is missing. Finally, I would recommend having scale bar numbers written on the image so readers can easily tell how small or big these granules without having to search through the figure legends.

4) The authors introduce several words that were not defined. For example: self/non-self-recognition, diffusion-entrapment mechanism, self-entrapment

Broader scope

1) The authors introduce the subject matter of RNA granules in lines 31-38 of how they are thought to form, liquid-liquid phase separation, without providing any indication of why these granules matter for biology. I think it will be more enticing for readers if it led to why you should care about granules. For example, RNP granules may compartmentalize biomolecules in distinct locations in the cell which may enhance certain biological interactions or reactions. Moreover, P bodies are thought to be important in stem cell differentiation, stress granules in cancer metastasis and virus infections, and germ granules in the specification of germ cell fate.  

Author Response

We would like to thank the Reviewer for their critical reading of our manuscript. We were thrilled to find out that the Reviewer indicated that our manuscript brings a much needed perspective of RNA granules to the field. As we indicated in the Abstract, to date the filed has mainly focused on the protein components of these granules despite the fact that these condensates are thought to primarily regulate the RNA biology of its constituents. They are called RNA granules after all. However, we also recognize that our manuscript needed clarifications, edits and some rewriting, to avoid inaccuracies and increase readability of our work. We have read Reviewer’s suggestions and comments carefully. After thorough discussion we agree with most of the Reviewer’s comments and implemented them where we could. All the edits in the manuscript are marked in red. With these edits, we believe that our manuscript emerged substantially stronger. Below, we address the Reviewer’s comments point-by-point.

Reviewer 2 Report

This review submitted by Tian et al. studies the RNA granules in different systems, from an RNA-centric point of view. They first explain the different RNA granules and their functions. Then, they explain role of the RNA molecules and the mechanisms that they play to induce the formation of such RNA granules.

In general, this review is well written and comprehensible. It is specially interesting the RNA focus in the formation of the RNA granules. However, the following remarks should be addressed:

  1. When describing the RNA granules, the authors only focus on three of them (P-bodies, stress granules and germ granules), however, later in the paper, they cover other granule types too, such as the neuronal granules. I consider that the paper would be benefited of introducing them before, to establish the differences among all of them. In that sense, the sudden appearance of nuclear RNA inclusions at the end of the manuscript is also a bit "out of context".
  2. The colors selected for the figure 1A are a bit unfortunate, as magenta+green=white. I understand the objective of the image, but as it is shown as a single panel, it is not clear whether all the white dots are only due to the mRNA molecules or whether there is a certain colocalization between DDX6 and G3BP1, or both.
  3. There are several sentences where the reviewer considers that the verb "enrich" should be used in passive, instead of the active form. For instance, sentences number 58, 159 or 303.
  4. Sentence 279: synapses should be spines
  5. Sentence 281: mechanisms should be mechanism
  6. Sentence 331: Which 3 types of granules do the authors refer to?
  7. Sentence 380: Figure 3D should be Figure 4.

Author Response

(The authors gave the same response as above.)

Round 2

Reviewer 1 Report

The authors have adequately addressed all my comments and misunderstandings. I have two minor points upon reading this new draft.  Looking forward to seeing it in print. 

1) On Line 43, the authors wrote “however, …”. I think However is unnecessary here because there is no contradiction with the connection between human diseases and RNA granules to the post-transcriptional regulation by RNA granules. 

2) On line 123, I recommend adding G3BP2 also as another key component in stress granules.